# Examining sex differences in neurodevelopmental and psychiatric genetic risk in anxiety and depression

**Joanna Martin** [1]*, **Kimiya Asjadi**[1], **Leon Hubbard**[1], **Kimberley Kendall** [1], **Antonio F. Pardiñas** [1], **Bradley Jermy** [2], **Cathryn M. Lewis** [2], **Bernhard T. Baune**[3,4,5], **Dorret I. Boomsma**[6], **Steven P. Hamilton**[7], **Susanne Lucae**[8], **Patrik K. Magnusson**[9], **Nicholas G. Martin** [10], **Andrew M. McIntosh** [11,12], **Divya Mehta**[13], **Ole Mors**[14,15], **Niamh Mullins**[16,17,18], **Brenda W. J. H. Penninx**[19], **Martin Preisig**[20], **Marcella Rietschel**[21], **Ian Jones** [1,22], **James T. R. Walters** [1,22], **Frances Rice** [1], **Anita Thapar** [1,22], **Michael O'Donovan** [1,22], **Major Depressive Disorder Working Group of the Psychiatric Genomics Consortium**[¶]

1 MRC Centre for Neuropsychiatric Genetics and Genomics, Division of Psychological Medicine and Clinical Neurosciences, Cardiff University, Cardiff, United Kingdom, 2 Social, Genetic and Developmental Psychiatry Centre, Institute of Psychiatry, Psychology & Neuroscience, King's College London, London, United Kingdom, 3 Department of Psychiatry, University of Münster, Münster, Nordrhein-Westfalen, Germany, 4 Department of Psychiatry, Melbourne Medical School, University of Melbourne, Melbourne, Australia, 5 Florey Institute for Neuroscience and Mental Health, University of Melbourne, Melbourne, Australia, 6 Dept. of Biological Psychology & EMGO+ Institute for Health and Care Research, Vrije Universiteit Amsterdam, Amsterdam, Netherland, 7 Psychiatry, Kaiser Permanente Northern California, San Francisco, California, United States of America, 8 Max Planck Institute of Psychiatry, Munich, Germany, 9 Department of Medical Epidemiology and Biostatistics, Karolinska Institutet, Stockholm, Sweden, 10 Genetics and Computational Biology, QIMR Berghofer Medical Research Institute, Brisbane, Queensland, Australia, 11 Division of Psychiatry, University of Edinburgh, Edinburgh, United Kingdom, 12 Centre for Cognitive Ageing and Cognitive Epidemiology, University of Edinburgh, Edinburgh, United Kingdom, 13 Centre for Genomics and Personalised Health, Faculty of Health, Queensland University of Technology (QUT), Kelvin Grove, Queensland, Australia, 14 iPSYCH, The Lundbeck Foundation Initiative for Integrative Psychiatric Research, Aarhus, Denmark, 15 Psychosis Research Unit, Aarhus University Hospital, Risskov, Aarhus, Denmark, 16 Social, Genetic and Developmental Psychiatry Centre, King's College London, London, United Kingdom, 17 Department of Genetics and Genomic Sciences, Icahn School of Medicine at Mount Sinai, New York, New York, United States of America, 18 Department of Psychiatry, Icahn School of Medicine at Mount Sinai, New York, New York, United States of America, 19 Department of Psychiatry, Vrije Universiteit Medical Center and GGZ inGeest, Amsterdam, Netherland, 20 Department of Psychiatry, Lausanne University Hospital and University of Lausanne, Lausanne, Switzerland, 21 Department of Genetic Epidemiology in Psychiatry, Central Institute of Mental Health, Medical Faculty Mannheim, Heidelberg University, Mannheim, Baden-Württemberg, Germany, 22 National Centre for Mental Health, Division of Psychological Medicine and Clinical Neurosciences, Cardiff University, Cardiff, United Kingdom

¶ Membership of the Major Depressive Disorder Working Group of the Psychiatric Genomics Consortium is listed in the S1 File.
* martinjm1@cardiff.ac.uk



**Data Availability Statement:** The data used in this study are third party data and were obtained via the method described below. There are ethical

## Abstract

Anxiety and depression are common mental health disorders and have a higher prevalence in females. They are modestly heritable, share genetic liability with other psychiatric disorders, and are highly heterogeneous. There is evidence that genetic liability to neurodevelopmental disorders, such as attention deficit hyperactivity disorder (ADHD) is associated with anxiety and depression, particularly in females. We investigated sex differences in family history for neurodevelopmental and psychiatric disorders and neurodevelopmental genetic

1 / 14

restrictions on sharing raw data (as the data include sensitive patient information) and as such the data are available upon request, as follows: NCMH data are available to External Collaborators via collaboration with NCMH Core Team members or Internal Collaborators. Further information can be obtained by contacting the NCMH Team: https://www.ncmh.info/contact/ PGC MDD data for secondary analyses are available via collaboration with PGC researchers, with details of the Data Access Committee and instructions for data access available here: https://www.med.unc.edu/pgc/shared-methods/data-access-portal/.

**Funding:** JM was supported by a Sêr Cymru II COFUND Fellowship from the Welsh Government (grant no. 663830 - CU189) and a NARSAD Young Investigator Grant from the Brain & Behavior Research Foundation (grant no. 27879). LH, AFP, and JTRW were supported by an MRC Mental Health Data Pathfinder grant (MC-PC-17212). CML and BJ are part-funded by the National Institute for Health Research (NIHR) Biomedical Research Centre at South London and Maudsley NHS Foundation Trust and King's College London. KK was funded by a Wellcome Trust Research Training Fellowship. The National Centre for Mental Health (NCMH) is funded by Welsh Government through Health and Care Research Wales (grant number: 514032). The PGC has received core funding from the US National Institute of Mental Health (5 U01MH109528-03). MOD, CML and the PGC are supported from the National Institute of Mental Health of the National Institutes of Health under Award Number U01MH109514. The content is solely the responsibility of the authors and does not necessarily represent the official views of the National Institutes of Health.

**Competing interests:** MOD, JTRW and IJ have received a collaborative research grant from Takeda Pharmaceuticals. Takeda played no part in the conception, design, implementation, or interpretation of this study. CML is on the Scientific Advisory Board of Myriad Neuroscience. All other authors report no conflicts of interest. This does not alter our adherence to PLOS ONE policies on sharing data and materials.

risk burden (indexed by ADHD polygenic risk scores (PRS) and rare copy number variants; CNVs) in individuals with anxiety and depression, also taking into account age at onset. We used two complementary datasets: 1) participants with a self-reported diagnosis of anxiety or depression (N = 4,178, 65.5% female; mean age = 41.5 years; N = 1,315 with genetic data) from the National Centre for Mental Health (NCMH) cohort and 2) a clinical sample of 13,273 (67.6% female; mean age = 45.2 years) patients with major depressive disorder (MDD) from the Psychiatric Genomics Consortium (PGC). We tested for sex differences in family history of psychiatric problems and presence of rare CNVs (neurodevelopmental and >500kb loci) in NCMH only and for sex differences in ADHD PRS in both datasets. In the NCMH cohort, females were more likely to report family history of neurodevelopmental and psychiatric disorders, but there were no robust sex differences in ADHD PRS or presence of rare CNVs. There was weak evidence of higher ADHD PRS in females compared to males in the PGC MDD sample, particularly in those with an early onset of MDD. These results do not provide strong evidence of sex differences in neurodevelopmental genetic risk burden in adults with anxiety and depression. This indicates that sex may not be a major index of neurodevelopmental genetic heterogeneity, that is captured by ADHD PRS and rare CNV burden, in adults with anxiety and depression.

# Introduction

Anxiety and depression are common mental health disorders, leading causes of distress and disability world-wide, and are associated with life-long adverse social, educational, and health outcomes, including premature mortality (e.g. from suicide) [1–3]. The aetiologies of these conditions involve a complex interplay of genetic and environmental risk factors and they are characterised by substantial clinical as well as aetiological heterogeneity [1, 3]. After puberty, anxiety and depression are each about twice as commonly diagnosed in females compared to males [1, 3, 4]. The reasons for these sex differences remain unknown. Twin studies suggest that depression may be more heritable in females, but the same has not been found for anxiety [5–7]. Examining whether risk factors have sex-specific impacts on these disorders has the potential to inform our understanding of aetiological heterogeneity and stratification of patients in order to aid clinical assessment and treatment.

Based on meta-analyses of twin and family data, mean heritability estimates are modest for diagnosed anxiety disorders (30–60%) and depression (37%) [7, 8], with similar estimates observed in males and females [7, 9]. Genome-wide association studies (GWAS) have identified robustly associated risk alleles for both disorders, accounting for 26% of variance in anxiety and 8.7% of variance in major depressive disorder (MDD) [10, 11]. The genetic correlation between males and females is very high for each of these disorders and larger samples are needed to identify SNPs showing a genome-wide significant sex difference in allele frequency [12, 13]. GWAS have also demonstrated that a significant proportion of the genetic liabilities of anxiety and MDD are shared with other psychiatric disorders in males and females [10–12, 14]. As such, considering sex-specific genetic effects that are shared across disorders is a complementary approach to primary genetic studies that examine sex differences in anxiety and depression.

Recent evidence points to the possibility that genetic risk factors associated with neurodevelopmental disorders, such as attention deficit hyperactivity disorder (ADHD) and autism,

which are more commonly diagnosed in males [15], may be more strongly associated with anxiety and depression in females compared to males. First, in children diagnosed with anxiety and/or depression, girls are more likely to carry a large, rare copy number variant (CNV) [16], a class of variants strongly linked to neurodevelopmental disorders [17–19]. Second, in an adult population (the UK Biobank), large, rare CNVs in loci previously associated with neurodevelopmental disorders have been associated more strongly with depression in women than in men [20]. Third, in children with anxiety or depression diagnoses, girls have a higher polygenic burden of common risk alleles for ADHD, compared to boys [21]. Finally, according to a Swedish register-based population study of children and adults, females with anxiety disorders are more likely to have a brother diagnosed with ADHD, compared to males with anxiety disorders [22].

However, inconsistent findings, that do not support a stronger association between neurodevelopmental genetic risk factors and anxiety and depression in females, have also been reported. The sex difference in ADHD polygenic burden in children has been observed for anxiety and depression based on nation-wide registers of clinical diagnoses, but this was not the case when diagnoses were based on research screening questionnaires in a study of the same population [21]. This discrepancy could indicate the influences of diagnostic biases, or an impact of severity of anxiety and depression. Depression also shows important developmental differences in aetiology and treatment [23] and no sex differences have been observed in the association between ADHD polygenic risk and anxiety or depression in older adults in the UK Biobank [24], suggesting that age at assessment may influence whether sex differences in liability to ADHD are observed in individuals with these disorders. Additionally, the Swedish register-based population study mentioned above found no sex difference in the rates of ADHD in the siblings of individuals with depression [22]. Taken together, the evidence for sex-specific manifestation of neurodevelopmental genetic risks comes from studies using register-based clinical diagnoses in children, with less consistent evidence in adults or using other methods of assessment of anxiety and depression.

Further work is needed to clarify the sex-specific impact of common and rare neurodevelopmental genetic risks on diagnosed anxiety and depression, using individuals ascertained from clinical mental health services. Previous studies using data from the UK Biobank rely on a relatively healthy, non-representative sample of the UK adult population [25]. Further investigation of sex differences is important to better understand anxiety and depression heterogeneity. Sex-specific manifestation of neurodevelopmental genetic liability as anxiety or depression in females could help explain under-diagnosis of neurodevelopmental disorders in females [15] and increased prevalence of anxiety and depression in females [1, 3], as well as poor treatment response to traditional anxiety and depression therapies in a proportion of those with depression and anxiety [26].

Another consideration is that of developmental differences, especially for depression. Depression in young people and adults do not show the same responses to treatment [27] and appear to show aetiological differences. Depression with an early onset (before mid 20s) shows higher heritability, stronger family history, and higher polygenic burden for schizophrenia and bipolar disorder, while even earlier onset depression (before adolescence) is associated with neurodevelopmental difficulties and ADHD risk alleles [28–32]. Some of these findings could be explained by retrospective recall in adult samples that could be associated with persistent depression, but on the whole, they suggest that age at onset is important to consider with regards to depression. Given the strong phenotypic and genetic relationship between depression and anxiety, it follows that age at onset should also be evaluated in the context of anxiety disorders.

The overall aim of this study was to examine sex differences in clinical and genetic profiles of individuals with anxiety and/or depression. Based on previous studies, we hypothesised that females would be more likely to have a family history of neurodevelopmental and psychiatric disorders and also a higher burden of genetic risk factors (common and rare) previously implicated in neurodevelopmental disorders, particularly in individuals with younger age at onset (before mid 20s) of their anxiety and depression.

## Materials & method

### NCMH community cohort sample

Volunteers were recruited from the National Centre for Mental Health (NCMH; https://www.ncmh.info/) research cohort, a Welsh Government-funded Biomedical Research Centre, operated by Cardiff, Swansea and Bangor Universities, in partnership with National Health Service (NHS) Health Boards across Wales [33]. NCMH collects biological samples, clinical data, and socioeconomic information from people across the lifespan with a history of neurodevelopmental, psychiatric, and neurodegenerative disorders and also from healthy control participants. Volunteers are recruited through several approaches, including via healthcare services, local advertisements, volunteer third-sector organisations, and by re-contacting participants from previous studies at Cardiff University. Informed written consent was obtained from adult participants. Written assent was obtained from children under 16 years old and written consent was obtained from a parent/carer. Approval for the NCMH was obtained from the Wales Research Ethics Committee and this specific study was also approved by the Cardiff University School of Medicine Research Ethics Committee.

Brief standardised interview assessments were completed by trained research assistants. Participants were asked about their personal and family history of mental illness, current medication use, and socioeconomic background. Information about psychiatric diagnoses was obtained via self-report, based on a list of conditions following the question: "Has a doctor or health professional ever told you that you have any of the following diagnoses?". For the current study, we included participants who self-reported receiving one or more diagnosis of any anxiety (including panic disorder, agoraphobia, generalised anxiety disorder (GAD), social phobia, and other unspecified anxiety disorder) or any depression (including single or recurrent major depressive disorder (MDD), mood disorder in pregnancy/postpartum, and other mood disorder not-otherwise-specified). For individuals <18 years old, parents/carers reported about their child's diagnoses. We excluded individuals with comorbid psychotic or neurodegenerative diagnoses and individuals aged >65 years old. We included a group of comparison individuals who reported no psychiatric disorders, psychiatric medication use, or family history of mental illness, to investigate whether anxiety/depression cases had an elevated neurodevelopmental and psychiatric genetic burden in the sample as a secondary analysis. The number of individuals meeting inclusion criteria was 4,178 with anxiety and/or depression and 157 comparison individuals.

We derived dichotomous socioeconomic variables as follows: a) low educational attainment was defined as leaving school without any GCSEs (UK school qualifications taken at 14–16 years of age at the end of compulsory education), b) low income was defined as a household income below £20,000/year, and c) no current occupation was defined as not being in education, employment or training. Participants reported on the age of first onset of any psychiatric symptoms, the age when they first came into contact with psychiatric services, and the age when they first received treatment for any psychiatric symptoms; however, specific information relating to the age at onset for anxiety/depression was not available. Information on presence of comorbid psychiatric disorders was available for the following: any

neurodevelopmental disorders (ADHD, autism spectrum disorder, developmental coordination disorder, tic disorders, intellectual disability, and dyslexia), obsessive compulsive disorder (OCD), post-traumatic stress disorder (PTSD), eating disorders, substance misuse and personality disorders. We also derived variables related to family history (1st or 2nd degree relatives) for neurodevelopmental disorders (ADHD, ASD or intellectual/learning disability), anxiety/depression, and schizophrenia/psychosis, as well as any of the above disorders. Participants were given a pack of standardised self-report questionnaires to return by post, which included the Hospital Anxiety and Depression Scale (HADS) [34]. The HADS is a well-validated instrument containing items relating to current (in the past week) symptoms of anxiety (7 items) and depression (7 items). Scores are summed to give dimensional measures of current symptom severity; data for each scale was considered missing if an individual was missing more than 1 item on the scale.

DNA samples were extracted from venous blood or saliva. The samples were genotyped using the Illumina PsychArray Beadchip, with rigorous quality control (QC) procedures; see details in S1 Text in S1 File. A total of 3,678,198 SNPs and 1,315 cases and 157 comparison individuals of European ancestry passed QC and were included in the analyses. Following QC, polygenic risk scores (PRS) were derived using common (>5% minor allele frequency; MAF), well-imputed (INFO>0.8) variants using PLINK version 1.9 [35], based on large discovery GWAS of primarily European ancestry, with no overlap with the target sample: ADHD (19,099 cases and 34,194 controls) [36], anxiety disorders (31,977 cases, 82,114 controls) [10], MDD (59,851 cases and 113,154 controls) [11], schizophrenia (67,390 cases and 94,015 controls) [37], ASD (18,382 cases, 27,969 controls) [38], and bipolar disorder (20,352 cases and 31,358 controls) [39]. For each discovery GWAS, PRS were calculated using 7 different p-value thresholds and the first principal component based on the correlation matrix for these PRS was extracted and analysed, using the PRS-PCA approach [40]; see details in S1 Text in S1 File. The PRS were standardised to be z-scores for each analysis.

CNVs were called using PennCNV [41] and CNV QC was conducted in samples that had passed the above SNP QC. See details in S1 Text in S1 File. Only rare (<1% frequency) variants of size >100kb passing QC were considered for analyses. A total of 1,056 cases and 139 comparison individuals passed CNV QC and were included in the analyses. Four dichotomous variables were derived for presence of: any neurodevelopmental CNV (based on a list of 54 CNVs [20]), any large (>500kb) CNV, as well as any large duplications and deletions separately.

## PGC clinical MDD sample

We also used a second dataset of individuals diagnosed with MDD from the Psychiatric Genomics Consortium (PGC) for ADHD PRS analyses; these data have been described elsewhere [11, 32]. Samples from 20 PGC contributing studies were available for analysis, including a total of N = 13,273 (67.6% female) MDD cases of European ancestry with genetic data available after all QC. Of these, 12,173 (91.7%) had available data on age-at-onset of MDD (see further details in S1 Text in S1 File). We calculated ADHD PRS in the cases from each of the 20 studies separately, following the same method as was used for the NCMH sample, described above, using common (>5% MAF), well-imputed variants.

## Analyses in the NCMH community cohort sample

First, we tested whether females were more likely to report family history of neurodevelopmental and psychiatric disorders.

To test whether females with anxiety/depression have a higher burden of common variant ADHD genetic liability than males, we tested for sex differences in ADHD PRS. As an

exploratory analysis, we also examined sex differences in other neurodevelopmental and psychiatric disorder PRS (i.e. ASD, schizophrenia, MDD, anxiety disorders and bipolar disorder). We used a conservative Bonferroni corrected p-value threshold of 0.01 for 5 tests for the exploratory PRS analyses.

To test whether age at onset impacts on association between PRS and sex, we restricted the sample to a group of individuals with earlier age at onset. As the samples contained few individuals younger than 18 years old (N = 14 in NCMH and N = 8 in PGC), we defined early age at onset as <26 years (comparable to previous work [32]). Although participants were asked to report on the age at first onset of their psychiatric problems, for individuals with comorbid problems, it was unclear if this related to anxiety/depression or a comorbid problem. Therefore, to define a group of individuals with early age-at-onset of anxiety/depression, we restricted the sample to those individuals who had anxiety and/or depression but no other reported comorbid psychiatric disorder (60.9% of the sample, 67.6% of whom are female), thereby excluding individuals who had comorbid neurodevelopmental disorders, oppositional defiant disorder, conduct disorder, eating disorders, OCD, PTSD, substance misuse, or personality disorders.

To test whether females with anxiety/depression have a higher burden of rare CNVs than males, we tested for sex differences in presence of neurodevelopmental CNVs, and any large (>500kb) CNVs, as well as large (>500kb) duplications and deletions separately.

In addition to testing our study hypotheses, we examined sex differences in clinical and socioeconomic characteristics of the sample and tested whether regardless of sex, individuals with anxiety and depression had higher neurodevelopmental and psychiatric PRS and were more likely to have rare CNVs relative to a comparison group.

For all analyses, males were coded as 0 and females were coded as 1, therefore OR>1 indicates a higher PRS or greater likelihood of CNV presence in females. Age at assessment was included as a covariate in all phenotypic analyses. The top 5 PCs and genotyping batch were included as covariates in all PRS and CNV analyses. Quality control metrics were included as covariates in the CNV analyses, to account for CNV calling quality: log R ratio (LRR) SD, B allele frequency (BAF) SD, and waviness factor (WF). The sample included full- and half-siblings. To account for related samples, we used a sandwich estimator to correct the standard errors of the regression coefficients. All analyses used generalised estimating equations (with a logit link) implemented in the *drgee* package in R [42]. Nagelkerke $R^2$ differences between null and full models were calculated to obtain estimates of variance explained. Analyses of count data with an expected cell count below 5 were performed using Fisher's exact test, although these results need to be interpreted cautiously as they were not corrected for covariates or related samples. Sensitivity analyses which included comorbid neurodevelopmental disorders as a covariate were performed, where appropriate.

## Analyses in the PGC clinical MDD sample

We first tested for sex differences in ADHD PRS in the whole sample of MDD cases. Next, we stratified the sample based on their age-at-onset (early onset: <26 years; later onset: >25 years) and re-ran the analysis in each of these groups. We also compared the early and later onset groups to each other. Analyses were run on each of the 20 studies separately and the results were meta-analysed using a fixed-effects model with inverse variance weights implemented in the *metafor* R package and weighted mean variance explained was calculated, using sample size for the weights. We calculated ancestry-based PCs for each of the 20 studies using PLINK, including the top 5 PCs as covariates in analyses.

## Results

### Description of the NCMH community cohort sample

The NCMH sample consisted of 3,032 individuals with any anxiety disorder diagnosis and 3,817 individuals with a diagnosis of any depression. In those with an anxiety disorder 88.1% had a diagnosis of depression, and 70.0% of those with a diagnosis of depression had an anxiety disorder. Given this high degree of comorbidity, we combined the two disorders into a primary sample of individuals with any anxiety and/or depression (N = 4,178; 65.5% female).

**S1 Table in** S1 File displays the clinical and socioeconomic characteristics of these individuals, stratified by sex. The age at assessment was 7–65 years old [mean(SE) = 41.5(0.21)], with 103 individuals aged <18 years old. Males were older at assessment and therefore age was included as a covariate in all phenotypic analyses. There were several other sex differences: males were older at reported onset of psychiatric symptoms, access to services and access to treatment, reported more severe current symptoms of depression and anxiety at assessment, were more likely to have comorbid neurodevelopmental disorders, PTSD and substance misuse, and had lower socioeconomic status (lower educational attainment and more likely to not be in education, employment or training). Females were more likely to have comorbid eating disorders and personality disorders. See **S1 Table in** S1 File for details.

### Family history & polygenic risk score analyses in NCMH community cohort sample

Females were more likely to report a family history of any major psychiatric disorder, and also of anxiety/depression, neurodevelopmental disorders, and psychosis/schizophrenia; see Table 1.

A sub-sample of N = 1,315 (63.2% female; age range: 11–65 years old; N = 14 individuals <18 years old) individuals with anxiety/depression had genetic data available after QC. Relative to the comparison group (N = 157; 56.1% female; age range 18–65 years old), individuals with anxiety/depression had higher PRS for anxiety, MDD, ADHD and ASD, but did not differ from the comparison group in terms of PRS for schizophrenia or bipolar disorder (see **S2 Table in** S1 File). The results for males and females separately were similar, with overlapping confidence intervals (see **S2 Table in** S1 File).

In the sample of those with any anxiety/depression, there were no sex differences in ADHD PRS; see Table 2. Exploratory analyses of other psychiatric disorder PRS (i.e. anxiety, ASD, bipolar disorder, MDD, and schizophrenia) also showed no sex differences (Table 2). Sensitivity analyses showed similar results after adjusting for comorbid neurodevelopmental disorders as a covariate (see **S3 Table in** S1 File). The sub-group of individuals with anxiety/depression with early age-at-onset and no comorbid disorders (N = 420) were more likely to be female (74.8% female) than those with later age-at-onset and no comorbid disorders (N = 299, 58.5%; p = 4.5x10$^{-6}$). In the early age-at-onset sub-group, males had higher PRS for anxiety and

**Table 1. Family history of psychiatric disorders of males and females with anxiety and depression in NCMH.**

| Phenotype | Males, N(%) | Females, N(%) | OR (95% CI) | P |
|---|---|---|---|---|
| **Family history of any psychiatric disorders** | 369(73.4) | 957(83.8) | 1.87 (1.45–2.40) | 1.4 x 10$^{-6}$ |
| **Family history of anxiety/depression** | 329(65.4) | 890(77.9) | 1.86 (1.47–2.34) | 1.6 x 10$^{-7}$ |
| **Family history of neurodevelopmental disorders** | 110(22.0) | 321(28.3) | 1.39 (1.08–1.78) | 9.6 x 10$^{-3}$ |
| **Family history of psychosis/schizophrenia** | 15(3.0) | 77(6.8) | 2.35 (1.33–4.14) | 3.1 x 10$^{-3}$ |

Males are coded as 0, females are coded as 1; therefore OR>1 indicates females have a higher reported family history. Age at assessment is included as a covariate.

**Table 2. Association of polygenic risk scores for ADHD (primary analysis) and other psychiatric disorders (exploratory analysis) with sex of individuals with anxiety and depression in a) the whole NCMH sample and b) the sub-group of NCMH individuals with early age at onset.**

| PRS | Whole sample (N = 1,315) | | | Early-onset sample (N = 420) | | |
|---|---|---|---|---|---|---|
| | OR (95% CI) | P | $R^2$ | OR (95% CI) | P | $R^2$ |
| ADHD | 1.03 (0.92–1.15) | 0.63 | $2.3 \times 10^{-4}$ | 1.14 (0.91–1.43) | 0.24 | $4.9 \times 10^{-3}$ |
| ANX | 0.94 (0.84–1.05) | 0.28 | $1.2 \times 10^{-3}$ | 0.75 (0.60–0.95) | 0.017 | 0.021 |
| ASD | 1.08 (0.96–1.21) | 0.21 | $1.7 \times 10^{-3}$ | 1.04 (0.84–1.30) | 0.73 | $4.1 \times 10^{-4}$ |
| BD | 1.01 (0.90–1.14) | 0.83 | $4.7 \times 10^{-5}$ | 0.92 (0.74–1.15) | 0.46 | $1.6 \times 10^{-3}$ |
| MDD | 1.01 (0.90–1.13) | 0.84 | $4.0 \times 10^{-5}$ | 0.92 (0.73–1.16) | 0.48 | $1.9 \times 10^{-3}$ |
| SCZ | 0.94 (0.84–1.05) | 0.29 | $1.1 \times 10^{-3}$ | 0.77 (0.61–0.96) | 0.020 | 0.019 |

ADHD: attention deficit hyperactivity disorder; ANX: anxiety disorders; ASD: autism spectrum disorder; BD: bipolar disorder; MDD: major depressive disorder; SCZ: schizophrenia. Males are coded as 0, females are coded as 1; therefore OR>1 indicates a higher PRS in females. Bonferroni corrected p-value threshold for exploratory analyses: p<0.01 (based on 0.05/5 tests).

schizophrenia compared to females (see Table 2), although these results did not withstand multiple testing correction.

### PRS analysis in PGC clinical MDD sample

The analyses testing for sex differences in ADHD PRS were also run using the PGC sample of individuals diagnosed with MDD (N = 13,273; 67.6% female). The mean age at assessment was 45.2 years old (SE = 0.12; range: 10–87 years). The results showed a weak sex difference in ADHD PRS, with slightly higher PRS in females compared to males [OR(CIs) = 1.06(1.02–1.10), p = 0.0033, $R^2$ = $2.3 \times 10^{-3}$]; see **S1 Fig in** S1 File.

Next, the sample was stratified into individuals who were diagnosed early (<26 years old; N = 6,215) and those who were diagnosed later (N = 5,852). The mean age at onset was 28.4 (SE = 0.13) years old. Across the 20 studies, females (mean(SE) = 27.6(0.16) years) were on average younger than males (mean(SE) = 30.1(0.23) years) at onset of their MDD (meta-analysis p = $6.5 \times 10^{-5}$). The observed sex difference in ADHD PRS was present in the early-onset group [OR(CIs) = 1.08(1.02–1.15), p = 0.0087, $R^2$ = $3.7 \times 10^{-3}$] and not significant in the later-onset group [OR(CIs) = 1.03(0.97–1.09), p = 0.34, $R^2$ = $3.1 \times 10^{-3}$]. However, the early- and later-onset groups did not differ in ADHD PRS [OR(CIs) = 0.95(0.90–1.00), p = 0.069, $R^2$ = $2.9 \times 10^{-3}$].

Given the nominally significant associations observed in the NCMH early onset group for anxiety and schizophrenia PRS, we tested these associations in the PGC data. The results were not replicated (see **S4 Table in** S1 File).

### Copy number variant analysis in the NCMH community cohort sample

In NCMH, a total of 1,056 cases (62.2% female) and 139 comparison individuals (56.8% female) were available for CNV analysis. The rate of large rare CNVs did not differ between cases (9.5%) and comparison individuals (10.1%) [OR(CIs) = 0.93(0.52–1.67), p = 0.82]. Only 2 comparison individuals had a neurodevelopmental CNV and there was no enrichment of such CNVs in cases [OR(CIs) = 2.20(0.55–19.20), p = 0.42]. Males and females with anxiety/depression did not differ in rate of neurodevelopmental or large CNVs, with similar results observed after adjusting for the higher prevalence of neurodevelopmental disorders in males; see Table 3. In the sample of individuals who were diagnosed early, the rate of any large (>500kb) CNV was also similar in males (N = 6, 7.1%) and females (N = 18, 7.2%), with no significant sex difference [OR(CIs) = 1.02(0.18–10.49), p = 1.00].

**Table 3. Association of copy number variants with sex of individuals diagnosed with anxiety/depression in NCMH individuals.**

| CNV category | Males N(%) | Females N(%) | OR (95% CI) | P | OR (95% CI)[a] | P[a] |
|:---:|:---:|:---:|:---:|:---:|:---:|:---:|
| Any | 42(10.5) | 58(8.8) | 0.78 (0.51–1.19) | 0.25 | 0.79 (0.51–1.23) | 0.30 |
| Dup | 33(8.3) | 43(6.5) | 0.71 (0.44–1.15) | 0.17 | 0.73 (0.44–1.22) | 0.23 |
| Del | 10(2.5) | 22(3.3) | 1.50 (0.66–3.41) | 0.33 | 1.51 (0.63–3.61) | 0.35 |
| ND | 15(3.8) | 18(2.7) | 0.74 (0.37–1.47) | 0.39 | 0.79 (0.39–1.58) | 0.50 |

[a] Analysis including presence of comorbid neurodevelopmental disorders as a covariate.

Dup: duplication; Del: deletion; ND: neurodevelopmental. Males are coded as 0, females are coded as 1; therefore OR>1 indicates a higher rate of CNVs in females.

## Discussion

Using data from the NCMH cohort, in individuals self-reporting clinical diagnoses of anxiety and depression, we find evidence of a higher rate of family history of neurodevelopmental and psychiatric disorders in females, but no evidence of sex differences in neurodevelopmental or psychiatric disorder polygenic risk or presence of rare CNVs. Using data from a clinical MDD sample from the PGC, we find weak evidence of higher ADHD polygenic risk in females compared to males. Overall, these results do not offer convincing support for the hypothesis that female anxiety and depression is more strongly associated with neurodevelopmental genetic risk burden compared to males.

The finding that females are more likely to report a family history of neurodevelopmental and psychiatric disorders is consistent with the Swedish register-based study of children and adults, which reported a higher rate of having a sibling diagnosed with ADHD in females with anxiety disorders, although the same sex difference in having a sibling with ADHD was not found for depression [22]. However, the results of our study may stem from factors independent of genetic burden. The NCMH sample is a volunteer cohort and is likely to be affected by sex differences in ascertainment, beyond the known observation that women are more likely to take part in research [25]. It is plausible that a family history of psychiatric problems may motivate females to seek clinical help earlier and to be more likely to take part in mental health research, compared to males. Females might also be more aware of and better able to recall the mental health difficulties of their relatives. Furthermore, family history of psychiatric disorders captures factors beyond genetic risk, including shared exposures and experiences (e.g. trauma or bereavement). We are unable to differentiate between these possibilities using our study design. The interpretation that the observed higher rate of family history in females is a true difference in underlying genetic burden is also not supported by the analyses of PRS and CNVs in the NCMH sample, which suggest that there are no clear sex differences in neurodevelopmental genetic burden in this sample.

The rate of any large (>500kb), rare CNVs in the NCMH data (cases + controls) was comparable though somewhat higher than in the UK Biobank (9.5% vs. 8.9%) [20] and the rate of neurodevelopmental CNVs was also higher in NCMH (2.9% vs. 1.2%), which could be partly explained by differences in sample ascertainment, the UK Biobank being a healthier sample on average. A recent UK Biobank study found that neurodevelopmental CNVs were implicated in depression, particularly in women, but did not find an overall association for any large (>500kb), rare CNVs. On the other hand, a Swedish study of children with anxiety/depression found a higher rate of large (>500kb), rare CNVs in girls [16]. In this study, we find no sex differences in presence of either neurodevelopmental or large (>500kb) CNVs in NCMH individuals with anxiety and depression, not supporting these previous findings. Although this could be a power issue, the sample size available for this analysis in our study is larger than that in the previous study [16] in children (N = 1,032 vs. N = 383).

We find mixed evidence of sex differences in ADHD PRS. In NCMH individuals with self-reported anxiety/depression, we observe no sex difference, but we do see higher ADHD PRS in females with MDD in the PGC data, in line with our hypothesis. However, the effect size for this association is small (OR = 1.06 per 1 SD increase in ADHD PRS). Exploratory analyses of other neurodevelopmental and psychiatric PRS in NCMH showed no robust sex differences. Although higher ADHD PRS have previously been reported in females with anxiety/depression in children [21], no sex difference has been found in adults [24]. The majority of the data used in this study came from adults (NCMH: 98.9% and PGC: 99.9%) and so we repeated the analyses in a sub-sample of individuals with relatively early age-at-onset (<26 years old). There was no sex difference in ADHD PRS in this sub-group in NCMH. The sex difference in ADHD PRS observed in the PGC MDD sample was present only in the early onset group, though this group did not differ from the later onset group in a direct comparison. While this analysis was not significant in the NCMH data, the point estimate was actually higher in the early onset group analysis in NCMH [OR = 1.17(0.93–1.47)] compared to PGC [OR = 1.08 (1.02–1.15)], indicating that the lower power of the smaller NCMH sample to detect such a small effect may have impacted on the lack of consistency in terms of statistical significance. This study also adds to other cross-disorder analyses of common genetic variants, which show moderate genetic correlations across ADHD, anxiety and depression, with no significant sex differences observed across these genetic correlations [12, 14].

Taken together, the results of this study provide mixed support that females with anxiety and depression may carry an increased burden of neurodevelopmental genetic liability compared to affected males. The analyses of an early onset subgroup, along with previous studies in children, indicate the possibility that there may be stronger sex-specific effects in diagnoses in children and young people.

The results need to be interpreted in light of the limitations of this study. We excluded individuals of non-European ancestries, due to methodological limitations of PRS analyses. The NCMH sample is a volunteer cohort of individuals mainly recruited from healthcare services across the UK and relied on self-reported diagnoses. Although there is evidence that even minimally phenotyped definitions of depression can be informative [43], the results may not generalise to stricter phenotypic definitions. In particular, strict diagnosis may capture more severe and recurrent depression, which is more heritable [8], and such differences in phenotypic definitions could explain the different results observed in NCMH and PGC. The NCMH sample may be less severely affected compared to some of the clinically-ascertained samples included in the PGC MDD sample. We observe a number of sex differences in socioeconomic and clinical variables in NCMH (**S1 Table in** S1 File), which indicate that males in the NCMH sample have lower educational attainment, and are less likely to be engaged in employment, education or training, and have more depression and anxiety symptoms. They are also older at assessment, symptom onset, access to services and obtaining treatment. Later age at onset and diagnosis in males is consistent with previous studies of depression [32, 44]. In general, these sex differences are expected given greater willingness to report emotional psychiatric symptoms and greater likelihood of participation in research by females [25, 45]. These sex differences also imply that, on average, females who have chosen to take part in the NCMH study have had their problems recognised and treated for longer and are less clinically impaired, compared to male study participants. Although based on the HADS scores, the NCMH sample is generally not a clinically severe group at the time of assessment, these sex differences in ascertainment could have resulted in under-sampling of phenotypically less impaired males. This could result in on average slightly higher genetic risk in the sampled males compared to a representative population of males with anxiety and depression, which could in turn have

reduced our ability to detect sex differences in ADHD genetic burden in the hypothesised direction.

The sample was cross-sectional and consisted primarily of adults, thus our definition of early onset included young adults (<26 years). However, due to the way this information was collected, we were only able to examine those who had anxiety and/or depression with no comorbid diagnoses (60.9% of the sample). Although this reduces the generalisability of the results to clinical populations with typical levels of comorbidity, it also means the results cannot be explained by any sex differences in patterns of comorbidity in the sample. We were also unable to stratify the sample based on comorbid ADHD and other neurodevelopmental disorders, due to the low sample size of reported comorbid diagnoses; adjusting for presence of neurodevelopmental disorders as a covariate had no discernible impact on the results.

One strength of the PGC sample is that it was larger than our primary sample and information on age at onset of MDD was available for the majority of the sample. However, the samples were not phenotypically the same, as the PGC samples had confirmed clinical diagnoses of MDD (and may or may not have had comorbid anxiety disorders), whereas the NCMH sample consisted of individuals with a variety of self-reported anxiety and/or depressive disorders, which may have increased the sample heterogeneity. Similar to the NCMH sample, the PGC MDD dataset was comprised primarily of adults, with only 8 individuals who were younger than 18 years old at assessment. Also, information on comorbid anxiety or neurodevelopmental disorders was not available in the PGC sample.

Finally, the PRS effect sizes and amount of variance explained in this study are small. Even in case-control studies, PRS for the same phenotype explain a small proportion of phenotypic variability and only a modest proportion of common genetic variation is shared across disorders [12, 14]. Therefore when examining cross-disorder shared genetic effects in relation to heterogeneity within clinically diagnosed samples the amount of phenotypic variance that can be explained is further reduced and greater power is needed.

In summary, the results of this study provide only weak evidence to support previous findings that anxiety and depression in females are associated with an increased neurodevelopmental genetic burden, compared to these diagnoses in males. In contrast to previous studies of children, there is less evidence for possible sex-specific cross-disorder genetic effects in adults with anxiety and depression. This suggests that sex is not a major index of cross-disorder genetic heterogeneity (in terms of neurodevelopmental genetic liability) in adults with anxiety and depression. Future research might benefit from focusing on child and adolescent populations.

## Supporting information

**S1 File.**
(DOCX)

## Acknowledgments

The National Centre for Mental Health (NCMH) is a collaboration between Cardiff, Swansea and Bangor Universities. We thank the NCMH study participants for their invaluable contribution to this project. We would also like to acknowledge the NCMH Team of research assistants, data managers (in particular Lawrence Raisanen), lab staff and principal investigators. This research was conducted using the mental health cross-disorder data resource, DRAGON-DATA, developed at Cardiff University.

We acknowledge the support of the Supercomputing Wales project.

With thanks to the Psychiatric Genomics Consortium Major Depressive Disorder Working Group for sharing the data for secondary analyses (see list of consortium members in S1 File). Lead author for this consortium group: Cathryn Lewis (Email: cathryn.lewis@kcl.ac.uk).

## Author Contributions

**Conceptualization:** Joanna Martin, Anita Thapar, Michael O'Donovan.

**Data curation:** Joanna Martin, Leon Hubbard, Kimberley Kendall, Antonio F. Pardiñas, Cathryn M. Lewis, Bernhard T. Baune, Dorret I. Boomsma, Steven P. Hamilton, Susanne Lucae, Patrik K. Magnusson, Nicholas G. Martin, Andrew M. McIntosh, Divya Mehta, Ole Mors, Niamh Mullins, Brenda W. J. H. Penninx, Martin Preisig, Marcella Rietschel.

**Formal analysis:** Joanna Martin, Kimiya Asjadi, Leon Hubbard, Kimberley Kendall, Antonio F. Pardiñas.

**Funding acquisition:** Joanna Martin, Ian Jones, James T. R. Walters.

**Supervision:** Anita Thapar, Michael O'Donovan.

**Writing – original draft:** Joanna Martin.

**Writing – review & editing:** Joanna Martin, Kimiya Asjadi, Leon Hubbard, Kimberley Kendall, Antonio F. Pardiñas, Bradley Jermy, Cathryn M. Lewis, Ian Jones, James T. R. Walters, Frances Rice, Anita Thapar, Michael O'Donovan.

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
