## [Decision Letter · Decision Letter 0]

29 Apr 2021

PONE-D-21-07612

Examining sex differences in neurodevelopmental and psychiatric genetic risk in anxiety and depression

PLOS ONE

Dear Dr. Martin,

Thank you for submitting your manuscript to PLOS ONE. After careful consideration, we feel that it has merit but does not fully meet PLOS ONE’s publication criteria as it currently stands. Therefore, we invite you to submit a revised version of the manuscript that addresses the points raised during the review process.

We look forward to receiving your revised manuscript.

Kind regards,

Sinan Guloksuz, M.D., Ph.D.

Academic Editor

PLOS ONE

Journal Requirements:

Please provide additional details regarding participant consent. In the ethics statement in the Methods and online submission information, please ensure that you have specified what type you obtained (for instance, written or verbal, and if verbal, how it was documented and witnessed). If your study included minors, state whether you obtained consent from parents or guardians. If the need for consent was waived by the ethics committee, please include this information.

Thank you for stating the following in the Competing Interests section:

MOD, JTRW and IJ have received a collaborative research grant from Takeda Pharmaceuticals. Takeda played no part in the conception, design, implementation, or interpretation of this study. CML is on the Scientific Advisory Board of Myriad Neuroscience. All other authors report no conflicts of interest.

We note that you have indicated that data from this study are available upon request. PLOS only allows data to be available upon request if there are legal or ethical restrictions on sharing data publicly. For information on unacceptable data access restrictions, please see http://journals.plos.org/plosone/s/data-availability#loc-unacceptable-data-access-restrictions.

4a) If there are ethical or legal restrictions on sharing a de-identified data set, please explain them in detail (e.g., data contain potentially identifying or sensitive patient information) and who has imposed them (e.g., an ethics committee). Please also provide contact information for a data access committee, ethics committee, or other institutional body to which data requests may be sent.

4b) If there are no restrictions, please upload the minimal anonymized data set necessary to replicate your study findings as either Supporting Information files or to a stable, public repository and provide us with the relevant URLs, DOIs, or accession numbers. Please see http://www.bmj.com/content/340/bmj.c181.long for guidelines on how to de-identify and prepare clinical data for publication. For a list of acceptable repositories, please see http://journals.plos.org/plosone/s/data-availability#loc-recommended-repositories.

One of the noted authors is a group or consortium Major Depressive Disorder Working Group of the Psychiatric Genomics Consortium. In addition to naming the author group, please list the individual authors and affiliations within this group in the acknowledgments section of your manuscript. Please also indicate clearly a lead author for this group along with a contact email address.

Please include captions for your Supporting Information files at the end of your manuscript, and update any in-text citations to match accordingly. Please see our Supporting Information guidelines for more information: http://journals.plos.org/plosone/s/supporting-information.

Reviewers' comments:

Reviewer's Responses to Questions

**Comments to the Author**

1. Is the manuscript technically sound, and do the data support the conclusions?

Reviewer #1: Yes

Reviewer #2: No

2. Has the statistical analysis been performed appropriately and rigorously? 

Reviewer #1: Yes

Reviewer #2: No

3. Have the authors made all data underlying the findings in their manuscript fully available?

Reviewer #1: No

Reviewer #2: Yes

4. Is the manuscript presented in an intelligible fashion and written in standard English?

Reviewer #1: Yes

Reviewer #2: Yes

5. Review Comments to the Author

Reviewer #1: Given the higher prevalence of anxiety and depression in female, the authors took efforts to investigated sex differences of genetic liability in individuals with anxiety or depression using both PRS and CNVs. The two datasets (NCMH and PGC) the authors selected to perform exploration and replicatory analysis were appropriated dataset with sufficient phenotypic information and statistic power. The results were very clear with absence of robust sex effects on PRSs and rare CNV burden. Only weak evidence of higher ADHD PRS in females compared to males were detected, particularly in early onset group. The results provide a better understanding of sex differences of anxiety and depression adults were absent in genetic heterogeneity for both PRS and CNV burden.

Introduction.

Since the authors has been described the heritability of anxiety and depression using family/ twin study. The sex effects in those equation models have been summarized and also valuable to introduce to the readers, e.g., PMID 26649194, 16212676, 14531579.

Methods.

PRS calculation: MHC region were excluded only when PRS-SCZ were calculated, why do not keep same protocol for other PRSs calculation? It is suggested to excluded more LD complex region (PMID: 18606306) to avoid PRS result bias.

PRSs were calculated for 1315 cases and 157 controls. The authors compared the PRSs only for cases between male and female after PRS standardized in total samples (including controls). As the different prevalence of cases in male compared in female, this standardization procedure may introduce bias for PRS sex comparison, which is suggested to excluded.

Multiple testing correction. In Table S2, only PRS-ADHD, PRS-ANX and PRS-MDD showed significant association with case control status. I personally suggest only compare these 3 PRSs in sex groups in following study.

Still in Table S2, do PRSs contribute the variance of case-control status differently between male and female? It is good to show the summary statistics separately in male and female group also in table S2.

Genetic PCs, please explain the reason that authors select first 5 PCs out of 20 PCs as covariates into the models.

Authors compared the sex difference of PRS and CNV burden separately. What if the comparison took both into account (e.g., Add CNV burden as covariates when comparing the PRSs)?

Results.

Table 2: the sex distribution is clear in total sample (63.2%); however, I could not find the sex distribution of early-onset samples and late-onset samples. Does sex distribution significant different from early-onset samples and late-onset samples? Please indicate this.

Discussion.

“It is plausible that a family history of psychiatric problems may motivate females to seek clinical help earlier and to be more likely to take part in mental health research, compared to males. Females might also be more aware of and better able to recall the mental health difficulties of their relatives.” This bias could be corrected by parenting report instead of self -report. To support this hypothesis, Chen et al found equal genetic component but larger shared environmental components in female compared with male in self-reporting data; and the equal shared, and nonshared environmental factors for both female and males using parent-report data.

“Furthermore, family history of psychiatric disorders captures factors beyond genetic risk, including shared exposures and experiences (e.g., trauma or bereavement). We are unable to differentiate between these possibilities using our study design.” It would be great if authors consider other shared environmental factors given the possibilities of dataset; e.g., Smoking, alcohol comsuption and education attendance.

Reviewer #2: While there are a number of strengths to this study, there are a number of problems with the underlying premise and methods. Their premise is that women with depression and anxiety disorders will have greater genetic risk for neurodevelopmental disorders like ADHD and autism. This is surprising given that population and clinical studies have shown that people onsetting with major depression (clinically diagnosed) are less likely to have early cognitive deficits, deficits that are associated with neurodevelopmental disorders. In fact, boys/men have higher rates of ADHD, autism and other neurodevelopmental disorders. So, this reader is not sure from where their hypothesis emanated. Further, sex differences in depression emerge just post-puberty with young women onsetting at a greater rate than young men (and this holds across the lifespan as women have higher rates of depression than men). Finally, childhood onset depression does not show sex differences in incidence. (The age of <26 years as “early onset” does not reflect early onset.)

Perhaps the previous studies on this topic from which they drew their hypotheses are reflecting the fact that across all psychiatric disorders, women experience higher levels of depressive and anxiety symptoms, just like women in general in the population. This may be one reason why depressive/anxiety symptomatology is associated with many psychiatric disorders and by sex.

However, this is not telling us about major depression as a clinical diagnosis. It seems that the NCMH sample is more a reflection of symptomatology experienced rather than frank diagnoses. This is not a minor point - misclassification of disorder (based on self reports) can have different effects on findings than clinical diagnoses. The PGC data are a better reflection of the latter.

In addition, the men in their sample were significantly different clinicially - greater severity, higher rate of neurodevelopmental disorders, greater substance use disorders - characteristics we know reflect men with MDD. Again, not sure then from where the original hypothesis is emanating if men in the sample themselves have higher rates of neurodevelopmental diagnoses.

Are the authors suggesting that there may be higher genetic risk for neurodevelopmental disorders in women with depression that then are expressed as depression in women and not expressed in childhood as neurodevelopmental disorders? This seems convoluted.

Additional point to address in the Introduction that stated no one had tested for sex differences in SNPS across disorders: Comorbidity of psychiatric disorders is pervasive and sex differences in shared SNPs were addressed for depression, bipolar disorder and schizophrenia in a large genome wide study in a recent paper in Biological Psychiatry by Blokland et al.

6. PLOS authors have the option to publish the peer review history of their article (what does this mean?). If published, this will include your full peer review and any attached files.

Reviewer #1: **Yes: **Bochao Danae Lin

Reviewer #2: No

---

## [Author Response · Author response to Decision Letter 0]

25 May 2021

Please see Response to Reviewers document

---

## [Decision Letter · Decision Letter 1]

17 Jun 2021

PONE-D-21-07612R1

Examining sex differences in neurodevelopmental and psychiatric genetic risk in anxiety and depression

PLOS ONE

Dear Dr. Martin,

Thank you for submitting your manuscript to PLOS ONE. After careful consideration, we feel that it has merit but does not fully meet PLOS ONE’s publication criteria as it currently stands. Therefore, we invite you to submit a revised version of the manuscript that addresses the points raised during the review process.

ACADEMIC EDITOR: There's a minor comment that needs attention. 

We look forward to receiving your revised manuscript.

Kind regards,

Sinan Guloksuz, M.D., Ph.D.

Academic Editor

PLOS ONE

Journal Requirements:

Reviewers' comments:

Reviewer's Responses to Questions

**Comments to the Author**

1. If the authors have adequately addressed your comments raised in a previous round of review and you feel that this manuscript is now acceptable for publication, you may indicate that here to bypass the “Comments to the Author” section, enter your conflict of interest statement in the “Confidential to Editor” section, and submit your "Accept" recommendation.

Reviewer #1: All comments have been addressed

Reviewer #2: (No Response)

2. Is the manuscript technically sound, and do the data support the conclusions?

Reviewer #1: Yes

Reviewer #2: Yes

3. Has the statistical analysis been performed appropriately and rigorously? 

Reviewer #1: Yes

Reviewer #2: Yes

4. Have the authors made all data underlying the findings in their manuscript fully available?

Reviewer #1: Yes

Reviewer #2: Yes

5. Is the manuscript presented in an intelligible fashion and written in standard English?

Reviewer #1: Yes

Reviewer #2: Yes

6. Review Comments to the Author

Reviewer #1: Although the authors mentioned that "the population ancestry variation captured by PCs

beyond the top 5 PCs is minimal, reflecting the relatively small and ancestrally

homogeneous nature of that sample.", the fraction of variance explained by top 5 PCs still not given in the manuscript, it is recommended to report.

Thanks for the responses of the rest are clear.

Reviewer #2: Authors primarliy responded adequately to concerns.

One comment with respect to the diagnostic question of depression - The use of "self report of depressive symptomatology" as distinct from a clinical diagnosis of depression is not only a "minimally phenotyped definition" but may have genetic risk finding implications given that the latter is more likely to demonstrate genetic risk than the former. In fact, that may be one reason why the findings differ across samples. A comment as such is warranted or perhaps should be discussed with regard to implications for their findings.

One additional minor note - In the introduction, the comment about “no SNPS showing genome-wide sex differences” gives the impression of a complete lack of sex differences in autosomal SNPs when this is not the complete story with regard to schizophrenia and major depression in the recent literature.

7. PLOS authors have the option to publish the peer review history of their article (what does this mean?). If published, this will include your full peer review and any attached files.

Reviewer #1: **Yes: **Bochao Danae Lin

Reviewer #2: No

---

## [Editor Report · Decision Letter 2]

22 Jul 2021

Examining sex differences in neurodevelopmental and psychiatric genetic risk in anxiety and depression

PONE-D-21-07612R2

Dear Dr. Martin,

We’re pleased to inform you that your manuscript has been judged scientifically suitable for publication and will be formally accepted for publication once it meets all outstanding technical requirements.

Kind regards,

Sinan Guloksuz, M.D., Ph.D.

Academic Editor

PLOS ONE
---

## [Editor Report · Acceptance letter]

25 Aug 2021

PONE-D-21-07612R2 

Examining sex differences in neurodevelopmental and psychiatric genetic risk in anxiety and depression 

Dear Dr. Martin:

I'm pleased to inform you that your manuscript has been deemed suitable for publication in PLOS ONE. Congratulations! Your manuscript is now with our production department. 

Kind regards, 

on behalf of

Dr. Sinan Guloksuz 

Academic Editor

PLOS ONE